# Exploring the Regulatory Role of ncRNA in NAFLD: A Particular Focus on PPARs

**DOI:** 10.3390/cells11243959

**Published:** 2022-12-07

**Authors:** Anirban Goutam Mukherjee, Uddesh Ramesh Wanjari, Abilash Valsala Gopalakrishnan, Ramkumar Katturajan, Sandra Kannampuzha, Reshma Murali, Arunraj Namachivayam, Raja Ganesan, Kaviyarasi Renu, Abhijit Dey, Balachandar Vellingiri, Sabina Evan Prince

**Affiliations:** 1Department of Biomedical Sciences, School of Biosciences and Technology, Vellore Institute of Technology (VIT), Vellore 632014, Tamil Nadu, India; 2Institute for Liver and Digestive Diseases, Hallym University, Chuncheon 24252, Republic of Korea; 3Centre of Molecular Medicine and Diagnostics (COMManD), Department of Biochemistry, Saveetha Dental College & Hospitals, Saveetha Institute of Medical and Technical Sciences, Saveetha University, Chennai 600077, Tamil Nadu, India; 4Department of Life Sciences, Presidency University, Kolkata 700073, West Bengal, India; 5Stem Cell and Regenerative Medicine/Translational Research, Department of Zoology, School of Basic Sciences, Central University of Punjab (CUPB), Bathinda 151401, Punjab, India

**Keywords:** PPARs, NAFLD, ncRNA, liver, NASH

## Abstract

Liver diseases are responsible for global mortality and morbidity and are a significant cause of death worldwide. Consequently, the advancement of new liver disease targets is of great interest. Non-coding RNA (ncRNA), such as microRNA (miRNA) and long ncRNA (lncRNA), has been proven to play a significant role in the pathogenesis of virtually all acute and chronic liver disorders. Recent studies demonstrated the medical applications of miRNA in various phases of hepatic pathology. PPARs play a major role in regulating many signaling pathways involved in various metabolic disorders. Non-alcoholic fatty liver disease (NAFLD) is the most prevalent form of chronic liver disease in the world, encompassing a spectrum spanning from mild steatosis to severe non-alcoholic steatohepatitis (NASH). PPARs were found to be one of the major regulators in the progression of NAFLD. There is no recognized treatment for NAFLD, even though numerous clinical trials are now underway. NAFLD is a major risk factor for developing hepatocellular carcinoma (HCC), and its frequency increases as obesity and diabetes become more prevalent. Reprogramming anti-diabetic and anti-obesity drugs is an effective therapy option for NAFLD and NASH. Several studies have also focused on the role of ncRNAs in the pathophysiology of NAFLD. The regulatory effects of these ncRNAs make them a primary target for treatments and as early biomarkers. In this study, the main focus will be to understand the regulation of PPARs through ncRNAs and their role in NAFLD.

## 1. Introduction

Non-alcoholic fatty liver disease (NAFLD) already affects 25% of the worldwide adult population and is steadily increasing [1]. Hepatic steatosis without a high alcohol intake is defined as NAFLD. Although there is undoubtedly a genetic component to NAFLD susceptibility, increased insulin resistance is strongly linked to the onset and development of this disease. It includes many liver diseases, including steatosis with inflammation through NASH, fibrosis, cirrhosis, and hepatocellular cancer [2]. Insulin resistance is the pathophysiological hallmark of the metabolic syndrome, and NAFLD is regarded as its hepatic component [3]. The “two-hit” concept, where steatosis is considered the first hit and oxidative stress and injury as the second hit, can explain the process of NAFLD [4]. Based on the underlying pathogenesis, NAFLD can be classified as a primary or secondary metabolic syndrome, and insulin resistance is linked to primary NAFLD. The processes of advancement from steatosis to more severe liver inflammation and fibrosis have been explained in recent years, given that the pathophysiology of NAFLD/NASH is poorly understood [5].

The processes of advancement from steatosis to more severe liver inflammation and fibrosis have been explained in recent years, even though the pathophysiology of NAFLD/NASH is still poorly understood. Steatosis is considered the first setting for the progression of NASH, although the second hit is necessary to recruit inflammation [6]. Hepatic steatosis mainly results from the overaccumulation of triglyceride (TG) fat vacuoles in the liver due to improper lipid metabolism. Compared to steatosis patients and healthy controls, hepatic CYP2E1 expression levels were elevated in NASH patients, and expression was localized in steatosis regions [7]. Another important event in steatosis and NASH may be an increased FAO in cytochromes, the increased ROS generated by the CYP enzymes enhancing hepatic oxidative stress, and worsening liver damage [8]. Another factor that increases hepatic steatosis is de novo lipogenesis. It has been observed that plasma glucose and insulin concentrations lead to de novo lipogenesis. It ultimately dysregulated the β-oxidation process, leading to hepatic steatosis’s progression [9]. During NAFLD progression, several factors, such as gut-derived microbial components, cytokines, and lipotoxicity, arise from different tissues, including adipose tissues [10]. Inflammatory responses in the adipose tissues eventually result in elevated levels of several pro-inflammatory cytokines, TNF, IL-6, IL-1, and CCL2, which are responsible for insulin resistance [10].

Additionally, endocrine conditions such as polycystic ovarian disorders, hypothyroidism, and growth hormone deficit may exist with NAFLD. The underlying endocrine pathology affects the treatment, which could change the prognosis of NAFLD [11]. According to recent studies, dehydroepiandrosterone sulfate insufficiency has been linked to the emergence of advanced NASH [12]. A bidirectional link connecting liver and endocrine processes is also shown by the growing reports of adrenal failure in individuals with end-stage liver disease and in those who have had liver transplants [13]. Endocrine dysfunction may result in metabolic liver disease in some people who may later be diagnosed with cryptogenic cirrhosis because endocrine hormones regulate cellular functions and the distribution of body fat [13]. ncRNAs were discovered in endocrine organs, and their functions in the growth and functions in endocrine tissues, as well as the possible associations of lncRNAs with particular disorders such as diabetes mellitus, were explained [14].

Contemporary living’s high-power consumption and sedentary habits, which fuel several harmful lifestyle-related disorders, particularly NAFLD, are reflected in the rising incidence rate [15]. The second reason for liver transplantation in the USA is NAFLD, an epidemic liver disease that affects roughly one-fourth of the global total [16]. Additionally, although around 20–30% of patients with normal weight have NAFLD, 75–100% of people with obesity have the disease [17]. Insulin resistance, dyslipidemia, and a pro-inflammatory state are the defining features of NAFLD. The core of current NAFLD treatment is weight loss through lifestyle adjustment, which is challenging for most patients to accomplish and maintain [18]. Few pharmacological alternatives are available, and patients with progressing NASH have generally been the focus of treatment [19]. PPARs regulate dyslipidemia, inflammation, atherogenesis, and glucose homeostasis (insulin-sensitizing characteristics) [20]. As a result, these substances should have diverse effects on NAFLD etiology, making them a prime candidate for therapeutic development. 

PPARs are a subclass of nuclear receptors known as peroxisome proliferator-activated receptors [21,22,23]. The nuclear receptor superfamily, of which PPARs are a part, has been shown to govern distinct stages of NAFLD through its many subtypes, making it an appealing target for therapeutic intervention [24]. For several metabolic illnesses, including diabetes, dyslipidemia, and cardiovascular diseases, including NAFLD, PPARs are thought to be potential therapeutic targets [25]. The activation of hepatic stellate cells (HSCs), inflammation, and glucolipid metabolism intimately associated with NAFLD are regulated by all PPARs isotypes [26]. The nuclear receptor family comprises PPARγ, PPARα, and PPARδ (also known as PPARβ) [27]. In adipocytes, PPARγ is highly expressed in the adipocytes and liver and acts as an inducer of adipocyte development [28]. PPARβ/δ is mainly expressed in skeletal muscles, with skin and adipose tissue expressing it to a lesser extent. Adipose tissue has high levels of PPARs, which are transcriptionally upregulated in steatosis, activate lipogenic enzymes, and worsen steatosis [29]. 

In the prognosis and progression of liver diseases, lncRNAs act as potential markers and serve as a direct target for therapeutic purposes. Numerous lncRNAs have been demonstrated to be related to liver disorders [30,31]. miRNAs are endogenous, tiny non-coding RNAs that play a crucial role in regulating the expression of mRNA and proteins of the target genes. The 3′-untranslated regions (3′-UTRs), which typically contain specific stability elements (including miRNAs binding sites), are the targets of the miRNAs at the post-transcriptional level [32,33,34]. In numerous pathophysiological processes, microRNAs play a crucial role as post-transcriptional regulators. Usually, they impact mRNA stability or translation, which represses the expression of the target gene. It is believed that microRNAs regulate gene expression in metabolic-related disorders, including NAFLD [35].

ncRNAs have emerged as important gene expression regulators at transcriptional, post-transcriptional, and post-translational levels. A plethora of research over the last several decades has provided evidence that ncRNAs coordinate most activities of liver metabolism. They are implicated in various illnesses and cellular processes, and there is evidence that they cooperate to form a dynamic regulatory network [36]. lncRNAs have been linked to various physiological and pathological processes, including cell growth, death, metastasis, angiogenesis, and liver function. LncRNAs, which act as drivers of NAFLD development, are overexpressed in NAFLD [37]. In this review, we will focus on the role of microRNAs, lncRNAs, and circular RNAs as functional coordinators of NAFLD progression, as well as the current understanding of how their dysregulation contributes to aberrant metabolism and pathology in insulin resistance and other associated features related to NAFLD. We will also focus on the role of PPARs in the ncRNA pathway regulation. With an emphasis on the relationship between ncRNAs and PPARs in the formation of NAFLD, we seek to present an overview of the interplay of these events in this review.

## 2. Activation and Regulation PPARs

PPARs regulate the homeostasis of lipids. There are generally three PPARs subtypes, known as PPARα, PPARγ, and PPARβ/δ [22]. The liver, which significantly impacts overall body nutrient/energy balance, is where PPARα governs lipid metabolism. PPARδ is also upregulated in oxidative tissues, affecting genes associated with substrate delivery, oxidative phosphorylation (OXPHOS), and energy homeostasis [27,38]. The stimulation of PPAR β/δ, which is crucial for regulating lipid and glucose metabolism in the liver, may slow the development of NAFLD [23,38,39,40,41].

### 2.1. PPARα

Although PPARs are distributed everywhere, the liver is where it is most prominently expressed. Fatty acid absorption, β-oxidation, ketogenesis, bile acid production, and TG turnover are all crucially regulated [42,43]. Due to its distinct roles in controlling lipid metabolism, encouraging TG oxidation, and suppressing hepatic inflammation, nuclear receptor PPARα has recently attracted special attention as a promising drug target for treating type 2 diabetes and associated illnesses. It has sparked interest in PPAR agonists as therapeutic targets for NAFLD [44]. 

### 2.2. PPARγ

The liver is significantly protected by PPARγ from oxidation, inflammation, fibrosis, fatty liver, and tumors [45]. Adipose tissue is where PPARγ is highly expressed and plays a crucial part in controlling adipocyte differentiation, adipogenesis, and lipid metabolism [46]. Additionally, it should be mentioned that NAFLD patients and experimental animals have dramatically increased hepatic PPARγ expression [42]. In addition to being closely associated with adipogenesis in mature adipocytes, PPARγ can enhance pre-adipocyte differentiation into mature adipocytes [47]. In dormant hepatic stellate cells (HSCs), PPARγ is highly expressed, however, PPARγ is repressed during the fibrosis process. According to studies, activating PPAR prevents HSCs from becoming active and lessens the amount of collagen deposited during hepatic fibrogenesis. PPARγ is, therefore, a valuable target in the treatment of fibrosis [48,49].

### 2.3. PPAR β/δ

Although PPARβ/δ is present in many tissues, alcoholic liver disease is also attributed to the noticeably increased expression of PPARβ/δ in the gut, liver, and keratinocytes [29]. Additionally, the connection between PPARβ/δ and hepatic lipid buildup in alcoholic liver disease has been researched [50]. Alcohol-induced hepatic lipid accumulation is suppressed by PPARβ/δ ligand activation [51]. PPARα also controls the production of carnitine-acylcarnitine translocase (CAC T) [52], a protein that aids in transporting fatty acids across the mitochondrial membrane [53]. Additionally, it can control how fatty acids are metabolized in peripheral tissues such as the liver. In line with these results, PPARβ/δ ablation worsens the hepatic TG response to alcohol consumption [54]. Additionally, numerous models prove that PPARβ/δ ligand activation blocks several molecular targets that stop NAFLD [55]. Additionally, various models suggest that PPARs may play a similar function in hepatotoxicity as a preventative and treatment for alcoholic liver disease (Figure 1) [56].

## 3. A Relationship between PPAR and NAFLD

The activation of total PPARγ can have detrimental effects by stimulating lipogenesis in the liver. However, there are advantages to partial PPARγ activation, mostly brought about by elevated adiponectin levels and lower insulin resistance. Adipocytes respond to PPARγ by secreting adiponectin, inhibiting resistin and visfatin, and differentiating into adipocytes [57]. Additionally, it blocks pro-inflammatory signaling molecules such as inducible nitric oxide synthase (iNOS), NF-κB, TNF-α, and IL-6, causing inflammatory cascades to be suppressed. The insulin-sensitizing activity of PPARγ ligands is mediated by all these effects [58]. 

Paradoxically, there is a strong correlation between the onset of NAFLD and hepatocyte-specific PPARγ expression. Numerous clinical research found that NAFLD patients had higher levels of hepatic PPARγ expression. Although PPARγ activation in adipose tissue offers a unique method for managing NAFLD, the side effect of activation of PPARγ will be hepatic steatosis [59]. Even though PPARγ’s primary function is in adipose tissue, where the highest expression levels are seen, patients with NAFLD have much higher hepatic expression levels of PPARγ, reflecting the fact that different tissues and cell types play different roles for PPARγ in different contexts. As a result, NAFLD patients using PPARγ agonists may see a different response in their liver than in their adipose tissue. Due to the various functions of PPAR-γ, new treatment strategies include the development of drugs that harness these positive effects while limiting adverse effects such as adipogenesis that can lead to weight gain. Additionally, preclinical research on dual agonists targeting two or more PPARs has yielded promising results, and some of these agents are now moving toward clinical trials [60]. 

Even though excessive adiposity and obesity are frequently linked to diabetes and insulin resistance, PPARγ works as an insulin sensitizer by encouraging the development of adipose tissue [61]. This apparent disparity is due to the development of metabolically abnormal adipocytes during chronic dyslipidemia, which is frequently accompanied by obesity. Hypertrophic adipocytes that produce TNF-α and have increased rates of lipolysis due to insulin resistance are characteristics of dysfunctional adipose tissue [62]. Enhancing lipolysis leads to the release of free fatty acids (FFA), which are ectopically stored in organs such as the liver, resulting in steatosis and lipotoxicity, ultimately leading to NASH and cirrhosis [63,64]. 

PPARγ has distinct functions in liver cells. PPARγ plays a steatogenic role in hepatocytes by regulating the expression of adipogenic genes, including Ap2 and CD36, which increases FFA uptake. The simultaneous activation of FAS and ACC1 facilitate the buildup of TG inside cells. It lowers the release of inflammatory cytokines (such as TNF-α and MCP1) and growth factors (such as TGFβ), which reduces inflammation and activates hepatic stellate cells, which in turn lessens fibrosis. A quiescent phenotype of the liver stellate cells (HSC) is also associated with PPAR-γ, limiting their activation and fibrosis [60]. 

The expected result of the elevated hepatic PPARγ expression observed in NAFLD patients would be activation of DNL in the liver cells and increased expression of adipogenic genes, exacerbating steatosis [65]. However, clinical trials suggest that the PPARγ agonists rosiglitazone and pioglitazone significantly reduce hepatic steatosis in NAFLD patients [66]. This mitigation is probably brought about by changes in the adipose tissue, where activation of PPARγ promotes the growth of healthy adipose tissue, limiting the shunting of extra lipids to the liver and the development of defective adipocytes [42]. The weight gain observed in TZD-treated NAFLD patients is consistent with an enhanced formation of adipose tissue [67]. Although liver PPARγ levels are elevated in NAFLD patients, the same factors that led to this upregulation throughout the illness are yet unknown. PPARγ may be activated in reaction to lipid build-up in hepatocytes, or it may be activated in response to stimuli before the build-up of lipids [68].

## 4. ncRNA Regulation in NAFLD

The molecular mechanisms underlying the initiation and development of NAFLD fibrosis are increasingly attributed to lncRNAs as significant contributors. There have been reports of specific potential processes by which lncRNAs may contribute to fibrogenesis, but much more research is still required to fully comprehend the role of lncRNAs in the progression of NAFLD fibrosis. Aptr, Malat1, Neat1, and HOTAIR are some of the lncRNAs found in the CCl_4_ mouse model that may also be relevant to human NAFLD fibrosis [69,70,71]. To understand the pathogenic mechanisms of NAFLD and establish miRNAs as diagnostic biomarkers, potential therapeutic targets at the early stage of NASH, and potential predictors of HCC, the emerging studies in NAFLD/NASH have used miRNA analysis as a starting point [72]. Several pieces of research on NASH have consistently found overexpression of miR-34a and downregulation of the liver-specific miR-122 [73,74]. CircRNAs_0046366 and 0046367 deficiency have been identified as NAFLD features, and restoration of these circRNAs reduces oxidative stress, lipotoxicity, and the severity of the illness. Circ_0071410 silencing has been demonstrated to reduce HSC activation, a crucial stage in developing liver cirrhosis. While circMTO1 adversely affects the progression of HCC, CDR1 and circ_0067934 can enhance the invasion and metastasis in HCC [75].

### 4.1. miRNAs

#### 4.1.1. miR-122

By increasing Sirt1 and turning on the AMPK pathway, miR-122 inhibition protects hepatocytes from lipid metabolic diseases such as NAFLD and inhibits lipogenesis [76]. By controlling its targets, such as FASN, ACC, SCD1, and SREBPs, miR-122 is crucial for the metabolism of fatty acids, TGs, and cholesterol, as well as for the terminal differentiation of hepatocytes [76,77,78,79]. Additionally, animals with mir-122a deletions triggered steatosis that resulted in NASH, fibrosis, and HCC, indicating that this miRNA is crucial for developing NAFLD. Mice lacking miR-122 had higher liver cholesterol and TG levels but lower serum cholesterol and TG levels [79]. Changes in serum and hepatic cholesterol and TG levels are caused by changes in very-low-density lipoprotein assembly and secretion in a way that is miR-122-dependent [80,81]. This disagreement has been seen in numerous research studies and may result from applying various models and inhibitory techniques [78]. Additionally, miR-122 participates in the NAFLD fibrogenic and carcinogenic-signaling pathways. miR-122 is a vital regulator of the epithelial-to-mesenchymal transition, a crucial step in developing chronic inflammation, fibrosis, and metastasis. When miR-122 expression was downregulated, MEKK-3, vimentin, and hypoxia-inducible factor-1 were activated [77,79,82].

#### 4.1.2. miR-21

According to Becker et al.’s analysis of the serum profiles of two cohorts consisting of 137 NAFLD/NASH patients and 61 healthy controls, the patients with NASH had higher levels of miR-21 in their blood than the NAFLD patients and healthy controls [83]. According to scientists, it may be caused by increased fibrosis in NASH patients [84,85].

#### 4.1.3. miR-34a

miR-34a is another miRNA that appears to play a role in the emergence of NAFLD. For instance, a study that involved 34 patients found that this miRNA was exclusively found in the serum of people with NAFLD/NASH and not in healthy individuals. This finding has also been verified by Liu et al. [86,87]. This abnormal increase has a deleterious influence on the signaling of fibroblast growth factors 19 and 21 and is strongly linked with BMI in patients with obesity [88,89]. Lipids also drive the expression of miR-34a, and since sirtuin 1 has been identified as one of its targets, this miRNA appears to have a role in exacerbating the symptoms of NAFLD and NASH, mostly by raising p53 acetylation and inducing hepatocyte death [79,90,91].

#### 4.1.4. miR-192

MiR-192 stands out among the miRNAs involved in NAFLD because it affects lipid synthesis by targeting stearoyl-CoA desaturase 1, making its upregulation a potential treatment for the condition [87,92,93,94,95].

#### 4.1.5. miR-370

The FA oxidation enzyme CPT1A is a direct target of miR-370. Fascinatingly, miR-370 may influence miR-122 expression, contributing to TG buildup in the liver. Additionally, miR-370 overexpression in HepG2 cells activates genes involved in lipogenesis, including FAS and ACC1, by altering the expression of SREBP-1c [35,96].

#### 4.1.6. Other miRNAs

By inhibiting cholesterol export and FA oxidation, miR-33 regulates cholesterol homeostasis [97]. Its blockage boosts reverse cholesterol transport, decreases LDL and TGs, and raises plasma HDH [98]. However, mice on a high-fat diet experience a faster progression of fatty liver and hepatic steatosis when miR-33 is knocked out. It is because miR-33 increases the number of lipids in the blood, particularly TGs [99,100]. According to prior studies, the upregulation of the miR-29 family inhibits insulin-stimulated glucose absorption through the Akt pathway by obstructing insulin signaling. In 2011, researchers found that the miR29 family is upregulated in HepG2 cells, causing insulin resistance that connects all components of the metabolic syndrome and is the most likely risk factor for developing NAFLD [101,102]. In the livers of humans and murine with NAFLD/NASH, the miR-873-5p increased, inhibiting GNMT. In in vivo NAFLD mouse models, its blockage decreases liver steatosis, inflammation, and fibrosis [103]. Several studies have demonstrated that miR-221 and its paralog miRNA-222 worsen hepatocarcinogenesis by targeting apoptosis-related proteins such as p53, p53 upregulated modulator of apoptosis (PUMA), NF-κB, and signal transducer and activator of transcription 3 (STAT3) [104,105].

Extracellular matrix (ECM) buildup in excess alters the liver’s typical architecture, causing pathophysiological damage that eventually leads to liver fibrosis–cirrhosis. ECM’s increased synthesis and deposition in response to a fibrotic stimulus have been linked to activated hepatic stellate cells (HSC) [106,107]. The main factor leading to liver fibrosis is the activation of HSCs. In stimulating LX-2 cells, miR-21 plays a vital role as a mediator through the PTEN/Akt pathway. miR-21’s fibrogenic effects on the activation of LX-2 cells are mediated. miR-21 may be a potential new molecular target for liver fibrosis [108]. The miR-27a and miR-199a are recognized as HSC activators during fibrogenesis, whereas miRNAs, including miR-335, miR-150, and miR-194, have been identified as HSC modulatory molecules [109]. However, more research is needed to determine their significance in fibrogenesis and to evaluate their relationship with TGF-β/Smad signaling [110]. A difference in HSC activation stages may cause a discrepancy between studies. An HSC is exposed to a variety of additional stimuli from surrounding cells. It may lead to in vivo activation of HSC with a different outcome. The expression and role of miR-192 in the activation of HSC in human livers with various pathologies, mouse livers with liver fibrosis, or HSC, must thus be further investigated [111].

### 4.2. lncRNAs

lncRNAs may serve as prospective indicators for the prognosis and development of liver disorders and direct treatment targets. Several lncRNAs have been linked to liver disorders. In this review, we will discuss different ncRNAs and their role in the progression of NAFLD. 

#### 4.2.1. MALAT1

The significance of lncRNA metastasis-associated lung adenocarcinoma transcript 1 (MALAT1) in increased proliferation and inflammation has received the most attention in studying a wide range of illnesses [112]. It is related to lung, liver, heart, and kidney fibrosis and can control gene expression at several molecular levels [113]. According to research on NASH fibrosis, MALAT1 is essential in modulating the chemokine CXCL5 [114]. According to a study by Xiang et al., in NAFLD, MALAT1 expression increased in vitro and in vivo, and MALAT1 knockdown prevented FFA from causing hepatocytes to accumulate lipids. Additionally, ARNT’s interaction with the PPARs’ promoter could limit PPARs’ expression. PPARs knockdown reversed these phenomena, whereas MALAT1 knockdown greatly increased PPARs’ levels and decreased CD36 expression. By controlling the miR-206/ARNT axis, MALAT1 regulated PPARs/CD36-mediated hepatic lipid accumulation in NAFLD. Consequently, MALAT1/miR-206/ARNT may be a therapeutic target for NAFLD [115].

#### 4.2.2. HOTAIR

During liver fibrosis, lncRNA and HOTAIR inhibits the expression of PTEN in hepatic stellate cells and is involved in the dysregulation of liver lipids. In the study by Li et al., they looked at whether HOTAIR could be a possible mediator of PTEN downregulation and lipid accumulation in hepatic cells at the onset of NAFLD. In HepG2 cells, exposure to FFAs increased TG accumulation by markedly boosting HOTAIR expression and suppressing PTEN expression (both at mRNA and protein levels). After stopping the FFAs therapy, the impacts on HOTAIR and PTEN expressions disappeared. In HepG2 cells treated with FFAs, PTEN downregulation and TG accumulation were inhibited by SiRNA-mediated HOTAIR knockdown, and they were also blocked by CAPE (an NF-Bp65 inhibitor). In HepG2 cells, FFAs may cause the upregulation of HOTAIR, which is likely dependent on NF-kB signaling. It would reduce PTEN expression and encourage the accumulation of TGs [116]. Interestingly, lncRNA HOTAIR suppressed another lncRNA, MEG3, through epigenetic pathways in a study on fibrotic animals. The subsequent reduction in MEG3 expression and promotion of liver fibrosis resulted from these promoter changes [117,118].

#### 4.2.3. APTR

The lncRNA Alu-mediated p21 transcriptional regulator (APTR), which is putatively involved in liver fibrogenesis, is increased in fibrotic liver samples. By preventing TGF-β-dependent induction of α-SMA in vivo, the reduction in APTR prevents collagen formation [119,120]. In fibrosis patients and animal models, high expression of APTR was seen [119]. Extracellular matrix protein (ECM) buildup and HSC activation were prevented by silencing APTR expression [119,121]. APTR was overexpressed and activated the hepatic stellate cells in fibrotic liver tissues. The activation of HSCs in vitro was prevented, and the build-up of collagen in vivo was reduced by APTR knockdown. In primary HSCs, p21 siRNA1 reduced the effects of APTR knockdown on the cell cycle and cell proliferation [120]. In vivo COL1A accumulation was reduced, and in vitro HSC activation was inhibited by APTR knockdown. Finally, elevated levels of APTR in the serum of patients with liver cirrhosis point to APTR as a potential biomarker for the disease [119]. Additional research on APTR in sera from large cohorts will likely provide more insight into the significance and role of APTR in fibrosis linked to NAFLD [122].

#### 4.2.4. PVT1

It was shown that Plasmacytoma Variant Translocation 1 (PVT1), whose function was more prominent in different malignancies, played a role in the development of fibrotic liver tissues by suppressing the expression of PTCH1 and stimulating the Hedgehog pathway. These pathways are essential for liver fibrosis and collagen deposition [123]. Patients with NAFLD had elevated relative expression levels of the lncRNA-PVT1. Additionally, patients with complex cirrhosis and hepatocellular carcinoma (HCC) have considerably greater levels of lncRNAPVT1 in the advanced stages of NAFLD. Given this, the lncRNA-PVT1 levels might be a helpful diagnostic biomarker for identifying patients with advanced NAFLD stages [124]. miR-152 was a driver of EMT and HSC activation through the suppression of Patchd1 (PTCH1) methylation and activation of the Hedgehog pathway during the characterization of a putative signaling network [69].

#### 4.2.5. lncRNA COX2

lncRNAs are becoming recognized as important players in the molecular mechanisms underlying the development and progression of NAFLD fibrosis [69]. Numerous lncRNAs are expressed as a result of germline-encoded receptors such as the Toll-like receptors. One of these, lincRNA-Cox2, activates and suppresses various immunological gene subclasses [125]. These genes appear to be co-regulated because the most strongly stimulated lncRNAs tended to be found in chromosomal locations where immune gene expression was also elevated. One of the most strongly stimulated lncRNAs was lincRNA-Cox2, located close to the prostaglandin-endoperoxide synthase 2 (Ptgs2/Cox2) gene [125]. In CCl4-induced fibrotic mice, lncRNA-Cox2, a lncRNA close to Cox2/Ptgs2, was examined since there were indications that it was involved in regulating inflammatory genes [126].

#### 4.2.6. NEAT1

By controlling heterogeneous nuclear ribonucleoprotein A2, NEAT1 expression was elevated in HCC, whereas its knockdown decreased HCC cell proliferation, invasion, and migration [127]. When Neat1 was knocked down using an adenovirus, the CCl4-induced liver fibrosis in these animals was reduced. Neat1 expression was also higher in the entire livers and primary HSCs produced from CCl4-treated mice than in oil-fed controls [128]. By modulating the c-Jun/SREBP1c axis by sponging miR-139-5p, NEAT1 exacerbated the FFA-induced lipid accumulation in hepatocytes, suggesting its potential as a novel therapeutic target for NAFLD [114]. For the prevention and treatment of NAFLD, lncRNA NEAT1 may be a potential biological target. Further evidence that the control of lncRNA NEAT1 expression in NAFLD subjects is connected to liver function and lipid metabolism comes from the robust correlation between the lncRNA NEAT1 expression in the peripheral blood of NAFLD patients and the ALT, GGT, TC, and TG levels [129]. NEAT1 increases the development of hepatocellular carcinoma, NAFLD, and liver fibrosis while acting as a preventative in the pathogenesis of acute-on-chronic liver failure by suppressing the inflammatory response [130,131]. Adipogenesis has been reported to need miR-140, and NEAT1 contains a particular binding site for this miRNA. NEAT1′s expression and stability are improved due to the interaction between miR-140 and NEAT1. Additionally, research has shown that adipogenesis needs NEAT1 to be activated in a miR-140-dependent manner [132].

#### 4.2.7. SRA

The steroid receptor RNA activator (SRA) was initially reported as a lncRNA that increases the expression of nuclear receptors’ steroid-dependent genes [131]. A study looking at the involvement of SRA in NAFLD in SRA knockout mice showed that the lack of SRA upregulates the expression of hepatic ATGL. Additionally, SRA inhibits the expression of ATGL in hepatocytes, preventing FFA from being oxidized. Although forced SRA expression suppresses ATGL expression and FFA-oxidation, loss of SRA in primary hepatocytes or a hepatocyte cell line upregulates hepatocyte function. SRA blocks the forkhead box protein O1 (FoxO1) transcription factor’s ordinary inductive functions to limit ATGL promoter activity [133].

#### 4.2.8. UC372

One of the ultra-conserved lncRNAs, the ultra-conserved element (UC372), has 100% identity in the rat, mouse, and human genomes [134]. The upregulation of UC372 in mice with type 2 diabetes (db/db mice), mice on a high-fat diet (HFD), and NAFLD patients suggest that this lncRNA has a role in liver steatosis and fatty liver. The prevention of miR-195/miR-4668-related target genes, such as fatty acid synthase (FASN), acetyl-CoA carboxylase (ACC), stearoyl-CoA desaturase 1 (SCD1), and lipid uptake-related genes such as CD36, has been proposed as a mechanism by which UC372 causes hepatic steatosis and causes the buildup of hepatic lipids [135].

#### 4.2.9. lncARSR

A previous study suggests that lncARSR may be connected to hepatic steatosis. It is less apparent how lncARSR affects NAFLD. A prior study showed that over-expressing the lncARSR gene accelerated the formation of liver fat both in vivo and in culture, suggesting that lncARSR may play a role in NAFLD and serve as a potential new treatment target for the disease [136]. The studies showed that lncARSR silencing might treat NAFLD by shutting down the IRS2/AKT pathway through YAP1. YAP1 has already been seen to rise in cases of liver disease as the severity of the liver damage increases. Another study revealed that LATS2 controlled YAP1′s phosphorylation and regulation in NAFLD. It has been demonstrated that lncARSR is linked with YAP1 and encourages YAP import into the nucleus [31].

#### 4.2.10. APOA4-AS

A plasma lipoprotein called APOA4 is involved in the control of several metabolic processes, including lipid and glucose metabolism. In rats, the liver and small intestine manufacture APOA4 before secreting it into the blood [137]. An in vivo investigation has demonstrated that APOA4 increases hepatic TG output [137]. According to a recent study, the antisense lncRNA APOA4-AS, produced from the APOA4 gene’s reverse strand and partially overlaps with the gene’s 3 ends, is crucial in controlling the expression of the APOA4 gene. This correlation could be a therapeutic target for treating NAFLD-affected patients [76].

#### 4.2.11. lncRNA H19

A nearby reciprocally imprinted gene for Igf2 and H19 is found in a highly conserved gene cluster. Steatosis and NAFLD caused by a high-fat diet (HFD) increased H19 expression. H19 silencing lowered the lipid accumulation in hepatocytes, while H19 overexpression caused lipid accumulation and upregulated several genes involved in lipid metabolism. By upregulating MLXIPL/ChREBP and silencing, Mlxipl reduced H19-induced hepatic steatosis. H19 silencing decreased the mTORC1 signaling complex, upregulated by H19 overexpression in hepatocytes. H19 increased hepatic steatosis by upregulating mTORC1 and MLXIPL in hepatocytes, according to hepatocyte implantation experiments [138,139].

#### 4.2.12. lncRNA NONMMUT010685 and NONMMUT050689

LncRNA NONMMUT010685 was identified to control the XBP1 gene in a co-expression system. The unrelated process of typical liver fatty acid production needed XBP1, a crucial regulator of the unfolded protein response. XBP1 plays a major role in human dyslipidemias [140]. Patients with NASH who have inadequate protein degradation in response to XBP1 may be more likely to develop cirrhosis [141]. In the co-expression network of the lncRNA-mRNA study, it was discovered that the lncRNA NONMMUT050689 might control the RIPK1 gene. It is known that RIPK1′s kinase activity promotes RIPK3-mediated necroptosis and that RIPK1 is involved in the inflammatory and cell death pathways. During NASH, it was discovered that RIPK1 in hepatocytes prevented the advancement of liver fibrosis [99,142].

#### 4.2.13. Other lncRNAs

LncRNAs can be employed as prospective targets for the diagnosis and therapy of NAFLD even though the molecular mechanism of the disease has not yet been fully understood. Upregulation of the lncRNA Gm15622 in NAFLD caused by HFD was examined in a study that discovered that Gm15622 could sponge miR-742-3p, boosting the quantity of SREBP-1 protein, a transcription factor that controls the expression of genes that govern the production of fatty acids, lipids, and cholesterol. Gm15622 overexpression can increase SREBP-1c expression, encouraging hepatic fat build-up [143]. In addition to being a possible prognostic biomarker and therapeutic target for PC, RUNX1-IT1 is an essential oncogenic lncRNA that controls and recruits RUNX1 to trigger c-FOS expression [144]. Inflammation, fibrosis, and NASH activity scores were all substantially linked with RUNX1 expression in NASH patients. Studies suggest a correlation between the lncRNA RUNX1-IT1 and its role in NAFLD [145] (Table 1).

### 4.3. Circular RNAs

Circular RNA (circRNA), a family of ncRNAs, was formerly considered a nonfunctional result of mRNA splicing [153]. Nevertheless, the properties of circRNA, such as tissue and development-specific expression, enrichment of miRNA response element (MRE), and tolerance to both RNase R and RNA exonuclease, indicate its putative function in eukaryotic gene regulation [154]. circRNA is a kind of rediscovered endogenous ncRNA that forms a covalently closed continuous loop via a specific splicing process and is regarded as the predominant subtype in gene transcription [155]. It demonstrates an extensive array of physiological and pathological activity. It has been demonstrated that circRNAs play crucial implications in cancer biology, but their implications in NASH remain unknown. circRNA, which exhibits tissue and pathology-specific expression, has been elucidated to influence circRNA–miRNA interactions [156]. Hepatic steatosis is a miRNA-related degenerative disease characterized by TG buildup and lipid peroxidation, which develops into non-alcoholic steatohepatitis, liver fibrosis/cirrhosis, and even hepatocellular cancer [72,157].

#### 4.3.1. *circRNA_0046367 and circRNA_0046366/miR-34a/PPARα*

circRNA_0046367 normalization decreased miR-34a’s inhibition activity on PPARα via preventing the miRNA/mRNA interaction with miRNA response elements during hepatocellular steatosis in vivo and in vitro (MREs). A unique epigenetic mechanism behind hepatic steatosis and accompanying oxidative stress is represented by the circRNA_0046367/miR-34a/PPARα regulatory system. In contradiction to its expression being reduced during steatogenesis, normalization of circRNA_0046367 eliminates miR-34a-induced PPARα inhibition and hepatic steatosis (Figure 2) [158]. 

circRNA_0046366 exerts its influence mostly on metabolic processes, particularly lipid metabolism. Recent research identifies PPARα, an NR1C nuclear receptor subfamily transcription factor, as a direct target of miR-34a [120]. This type of liver-specific, ligand-activated PPARs isoform stimulates the expression of several genes with lipometabolic properties [159]. In obese patients, the absence of PPARα confers a high risk of insulin resistance, depletion of n-3 long-chain polyunsaturated fatty acids, and liver steatogenesis [160]. Accumulating data show a miR-34a/PPARα regulatory mechanism, which is the essential mediator of circRNA_0046366 effects on hepatic lipid metabolism (Figure 2). circRNA_0046366, whose expression was reduced in HepG2-based hepatocellular steatosis, inhibits the function of miR-34a. Silencing of miR-34a eliminates its inhibitory effect on PPARα and restores PPARα expression [161]. PPARα restoration enhances the expression of TG metabolism-related downstream genes [i.e., carnitine palmitoyl transferase 1A (CPT1A) and solute carrier family 27A (SLC27A)] at the transcriptional and translational levels. miR-34a is highly upregulated in multiple rodent hepatic steatosis models [162,163], and its expression persistently corresponds well with the clinical presentation of hepatic steatosis in Chinese [87], Japanese [164], and Philippine populations [165]. Targeted inhibition of PPARα underlines the steatosis-related action of miR-34a [120]. Targetome and targetome-based pathway analyses of miR-34a were merged to identify the miRNA-dependent downstream signaling and activities of circRNA_0046366 [166,167]. Using databases of ncRNA (circBase and miRBase) and aircRNA–miRNA interaction, algorithms to elucidate the circRNA–miRNA connection underpinning hepatic steatosis, expression patterns of circRNA_0046366 and miR-34a were examined in a HepG2-based steatosis model stimulated by high levels of fat [161].

#### 4.3.2. *circRNA_021412/miR-1972/LPIN1*

Guo et al., 2017, identified Lipin 1 (LPIN1) as a transcriptional regulator of circRNAs on metabolic pathways. The circRNA-miRNA-mRNA network subsequently revealed the signaling cascade of circRNA_021412/miR-1972/LPIN1, characterized by a lower level of circRNA _021412 and miR-1972-based regulation of LPIN1. Hepatic steatosis was caused by the LPIN1-induced suppression of long-chain acyl-CoA synthetases (ACSLs) expression [168]. 

LPIN1 could be the critical mediator of the transcriptional regulatory action of circRNAs on metabolic pathways. In the circRNA–miRNA–mRNA network, Has_circRNA_021412 and miR-1972 with LPIN1-regulatory activity and LPIN1 were identified as essential nodes. During hepatic steatosis, the interplay of the circRNA, miRNA, and LPIN1 exposes several unexpected yet crucial pathways [168]. 

LPIN1 may preferentially activate fatty acid oxidation and mitochondrial oxidative phosphorylation, as demonstrated by gain-of-function and loss-of-function techniques [169,170]. Furthermore, downregulated LPIN1 inhibits fatty acid breakdown in an ACSL-dependent manner. In addition, LPIN1 is confirmed to be the crucial element of multiple signaling transductions that are deeply involved in lipid homeostasis, such as SIRT1/AMPK signaling [171], mammalian target of rapamycin complex 1 (mTORC1)/SREBP signaling [172], NF-E2-related factor 1 (Nrf1) signaling [173], and hepatocyte nuclear factor 4 (HNF4) signaling [168,174].

#### 4.3.3. *circRNA_002581/miR-122/SLC1A5, PLP2, CPEB1*

circRNA has been shown to play a role in liver regeneration [175] and to serve as a molecular marker and therapeutic target for liver cancer [176]. miRNAs serve crucial roles in a variety of liver diseases [177]. MiR-122 levels were considerably reduced in NASH patients [178], whereas liver-specific and germline miR-122 deletion mice developed hepatic steatosis in early adulthood, followed by NASH, liver fibrosis, and eventually HCC as they aged [80]. 

circRNA_002581 (also known as circ_0001351 in circBase) is situated in the exonic region of the mouse chromosome 5 between positions 66,753,956 and 66,756,359, with a complete length of 2403 bp and a spliced length of 275 bp. The circRNA_002581–miR-122–CPEB1 pathway is implicated in NASH etiology [179,180]. By sponging miR-122, circRNA_002581 overexpression significantly alleviated the inhibitory effect of miR-122 on its target CPEB1. circRNA_002581 knockdown significantly attenuated lipid droplet accumulation, decreased alanine aminotransferase (ALT) levels, aspartate aminotransferase (AST), pro-inflammatory cytokines, apoptosis, and hydrogen peroxide, and raised the level of ATP in both mouse and cellular models of NASH (Figure 3) [179].

#### 4.3.4. *circRNA_0067835/miR-155/FOXO3a*

circRNAs serve primarily as miRNA sponges, binding to functional miRNAs and then regulating gene expression. By functioning as a sponge of miR155 to boost FOXO3a expression, circRNA_0067835 modulated hepatic fibrosis development, suggesting that it may offer targeted therapy for patients with liver fibrosis [181]. The FOXO family of Forkhead transcription factors has been discovered to regulate HSC proliferation via the PI3K/Akt pathway [182]. FOXO3a is essential for the integration of various pathways. FOXO3a proteins are crucial in maintaining intracellular redox equilibrium under diverse environmental stresses [183]. In the study by Lili zhu et al., bioinformatics analysis and subsequent luciferase reporter experiments were used to identify the FOXO3a gene as a direct target of miR-155. FOXO3a and AKT expression levels were considerably reduced in miR-155-overexpressing cells and elevated in miR-155-silenced cells, indicating that miR-155 modulated AKT/FOXO3a signaling (Figure 4). There are correlations between AKT/FOXO3a signaling and liver fibrosis since AKT/FOXO3a expression was considerably enhanced in HSCs and CCl4-treated liver [181]. LX 2 cells are the predominant cell type involved in liver fibrosis [157,160,161].

#### 4.3.5. *circRNA_0074410/miR-9-5p/KEGG Pathway*

The profiling of circRNAs in fibrotic HSCs indicated that 179 and 630 circRNAs were elevated. Recent research revealed that circ_0074410 inhibited the expression of miR-9-5p and increased HSC activation via the α-SMA protein. Suppression of hsa_circ_0071410 also increased the expression of miR-9-5p, resulting in a decrease in irradiation-induced HSC activity [122,184].

#### 4.3.6. *circRNA_34116/miR-22-3p/BMP7*

In a mouse model of liver fibrosis generated by CCl4, microarray analysis found 10,389 circRNAs, of which 69 were differentially expressed in fibrotic liver tissues; 55 were negatively regulated, and 14 were activated [185]. An in silico study anticipated the existence of miR-22 MRE on circRNA_34116 and suggested that this circRNA can link to miR-22-3p competitively and consequently influence the transcription of its target gene bone morphogenetic protein 7 (BMP7) [186]. In conclusion, networks between circRNAs and miRNAs have evolved as an entirely new strategy for gene expression regulation. They may facilitate the identification of new treatment targets and enhance our knowledge of the molecular regulation of disease advancement and progression [122].

## 5. ncRNA Regulation of PPARs in NAFLD

ncRNA may have significant regulatory functions in the initiation and progression of NAFLD. This class of molecules, including the lncRNA, circRNA, and miRNA, does not code for proteins but still affects gene expression [187]. miRNAs have many functions, including regulating posttranscriptional gene expression, adipocyte differentiation, lipid metabolism, cholesterol metabolism, insulin resistance, and immune responses [188,189]. lncRNAs are 200 nucleotides in size and act as a transcriptional regulators of gene activation or silencing through chromatin modification [190]. There are many conserved binding sites on circRNA, which function as a “miRNA sponge” by inhibiting miRNA activity by interacting with miRNA AGO proteins [191].

### 5.1. miRNAs

#### 5.1.1. miR-124-3p

According to accumulating data, the development and progression of NAFLD are attributed to miRNAs. However, little is known about the molecular model of miRNA in this NAFLD [122]. PPARα is known to be a target of many miR-124. A study has reported the role of miR-124-3p in the progression of NAFLD. The study showed that miR-124-3p targets Pref-1, the preadipocyte factor 1 [192]. During a high-fat diet, miR-124-3p activity in the liver rises. Suppressing the expression of miR-124-3p lowers hepatocyte lipid levels and inflammatory markers, and vice versa. As a result, miR-124-3p is an important regulator of the liver’s lipid homeostasis. It was also noted that when Pref-1 expression is inhibited in preadipocytes, it increases PPARs’ expression, and preadipocytes eventually differentiate into adipocytes [193]. 

Another study reported that when Pref-1 was downregulated, and PPARs were increased by blocking Notch signaling, it improved adipogenic differentiation [194,195]. It is consistent with lower PPAR activity, which has already been linked to decreased TGHs ATGL and CES1 expression [196].

#### 5.1.2. miR-21

Liver steatosis is one of the features found in NAFLD patients. miR-21 targets PPARs during the development of NAFLD [197]. In NASH patients and diseased mice, miR-21 levels were elevated in the liver and muscle, which reduced the PPAR-α expression [85]. An investigation that was recently completed revealed this observation. This study also showed that β-oxidation in hepatocytes is known to be controlled by PPARs, which also have anti-inflammatory effects on non-parenchymal cells. Consequently, it is possible that the pro-metabolic and anti-inflammatory activities of PPARs in various liver cells contributed partially to the enhanced behavior of miR-21 KO mice. The production of C/EBP and PPARs is suppressed by β-catenin when the classical WNT/β-catenin signaling pathway is activated, which prevents pre-adipocyte differentiation [198]. It was reported that by targeting low-density lipoprotein-related receptor 6 (LRP6) to promote the WNT signaling pathway, miR-21 production could be inhibited to treat NAFLD [199]. LRP6 is the target of miR-21, which controls the classic WNT signaling pathway in NAFLD. By regulating the miR-21-5p/SFRP5 pathway, PPARs were reported in another study as being able to reduce oxidative stress and inflammation in NASH, suggesting a potential designated target for NAFLD therapy [147].

#### 5.1.3. miR-122

Hepatic cholesterol, as well as the metabolism of lipids, are regulated by miR-122, considered the most prevalent miR in the liver [200]. Its role in NAFLD has also been reported and showed an increased expression pattern in the affected individuals [94]. It has been demonstrated that miR-122 is necessary for translating respiratory proteins, increasing the respiratory enzyme activity of mitochondria, and improving mitochondrial proteostasis in the liver [201]. Multiple signaling pathways are targeted by miR-122 to control lipid metabolism. FFAs increased the expression of miR-122, facilitated its release into the bloodstream, and inhibited TG production by targeting Agpat1 and Dgat1 and boosting β-oxidation. The scientists also discovered that blocking miR-122 caused FOXO1 to be downregulated and PPARs to be upregulated, showing that miR-122 is implicated in several pathways relevant to lipid metabolism [202,203].

#### 5.1.4. miR-34a

Several studies have reported the role of miR-34a in NAFLD and hepatic steatosis [163]. Numerous investigations found that people with NAFLD had greater levels of miR-34a. With increasing disease severity, miR-34a, apoptosis, and acetylated p53 levels grew [120]. Hepatic PPARs and SIRT1, which are miR-34a’s major targets, were significantly suppressed due to the upregulation of miR-34a. When miR-34a was silenced, the expression of PPARs, SIRT1 and PPAR’s downstream genes initially rose [147]. Additionally, there was a rise in AMPK activation, the primary metabolic sensor. The miR-34a inhibitor reduced the level of steatosis and reduced lipid build-up. A recent study investigated the role of the SIRT1 signaling pathway in miR-34a by inducing resveratrol in high-fat diet-fed rats [204].

#### 5.1.5. miR-130a and miR-130b

In a study conducted in 2017, it was reported that the hepatic expression of miR-130a-3p was significantly reduced in mouse models with fibrotic steatohepatitis. NASH fibrosis has a complicated pathophysiology. There are few effective treatments, and researchers do not fully comprehend how liver fibrosis works. miRNAs are crucially implicated in the many phases of liver fibrosis, particularly HSC activation, proliferation, and ECM synthesis [205]. HSC apoptosis is a crucial component of NASH-related liver fibrosis that could be associated with the severity of NAFLD. Increased production of the initiator caspase-9 and effector caspase-3, which miR-130a-3p triggers, causes the proteolytic cleavage of PARP, which results in cellular disintegration and apoptosis [206]. In a study conducted in rat HSC-T6 cells, miR-130a and miR-130b reduced PPAR expression by specifically binding to the 3′-UTR of PPARs mRNA. In cell culture, miR-130a and miR-130b overexpression, PPARs’ downregulation, and ECM gene overexpression may be mediated by TGF-1. These results imply that PPARs are downregulated in liver fibrosis by miR-130a and miR-130b [207]. In another study completed in HepG2 cell lines and primary mouse hepatocytes, miR-130a-3p overexpression enhances insulin signaling, whereas miR-130a-3p silencing has the opposite effect. MiR-130a-5p has little impact on the control of insulin signaling [208]. It has been demonstrated that miR-130a-3p effectively inhibited the production of PPARs-regulated genes by targeting the mRNA coding and 3′UTRs. It is believed that miR-130-3p effects on (human fatty acid synthase) FAS expression may be indirectly mediated by reducing PPARs expression since PPARs might control FAS expression [208], thus, playing a role in NAFLD.

#### 5.1.6. miR-155

The metabolism of lipids plays a major role in the pathogenesis of NAFLD. C/EBP, SIRT1, and PPARs are considered to be targets of miR-155. PPAR α promotes utilization in regulating fat metabolism, whereas PPARγ activation increases storage [209]. They are also known to act as an anti-fibrotic gene which is a direct target of miR-155 [210]. Variations in steatosis, inflammation and fibrosis are regulated by signals in normal lipid and insulin regulation and the control of inflammatory responses due to miR-155′s effect on PPARs signaling. It is possible that miR-155 functions as a negative regulator of adipogenesis because the miR-155 expression is lowered during in vitro adipogenesis and miR-155 overexpression inhibits the production of PPARs and cEBP (Figure 5) [211].

### 5.2. lncRNAs

#### 5.2.1. lncRNA FLRL6/FLRL2

In NAFLD, it was noted that seven lncRNAs and five mRNAs expression patterns were associated with circadian rhythm. In the pathway analysis, the central molecule in the circadian rhythm, Per2, was found to be a target of lncRNA FLRL6, and it was also observed that the level of this lncRNA increased about 3.3-fold along with which the level of Per2 was also elevated by about 3.5-fold [212,213]. A study reported that FLR2 lncRNA, an upstream lncRNA of another circadian-rhythm target aryl hydrocarbon-receptor nuclear translocator-like (ARNTL), was downregulated in the rodent models affected with NAFLD. In the previous study, several lncRNAs, including FLRL8, FLRL3, and FLRL7, were associated with lipogenesis via proteins in the PPARs signaling pathway, suggesting their potential regulatory involvement in lipid metabolism [212].

#### 5.2.2. lncRNA H19

One of the first and most extensively studied lncRNAs in liver illness is H19. Hepatocellular carcinoma patients had higher levels of H19, which inhibited tumor metastasis in a miR-220-dependent manner [214]. In in vitro and in vivo models of NAFLD study, H19 has a molecular function in controlling NAFLD by directly modulating the miR-130a/PPARs axis, inhibiting hepatic lipogenesis [152]. Fatty acids elevate the expression of H19 in hepatocytes and diet-induced fatty liver, and H19 overexpression might promote steatosis and increase lipid accumulation. Obstructive cholestatic liver fibrosis quickly developed due to hepatic overexpression of H19 [215]. These suggested that H19 could play a crucial role in NAFLD and act as a lipid sensor to regulate hepatic metabolic balance.

#### 5.2.3. MALAT 1

In vivo and in vitro studies have shown the role of lncRNA MALAT-1 in NAFLD. Patients with NAFLD were observed to have elevated MALAT1, which increased the development of liver fibrosis [216]. According to previous research and bioinformatics analysis, a correlation between the binding site of miR-206 and MALAT1 or aryl hydrocarbon-receptor nuclear translocator (ARNT) was found [115]. More studies are required to thoroughly analyze and study the exact involvement of PPARs in NAFLD by regulating lncRNA MALAT 1.

### 5.3. CircRNAs

A study noted that circRNA expression patterns in developing bovine adipose tissue in animals showed that circFUT10 overexpression strongly reduces PPARs and C/EBPa expression [217]. By targeting PPARs, GC1B, circFUT10 acts as a ceRNA for miRNA let-7c/let-e to control the differentiation of bovine adipocytes. Adipocyte differentiation and tissue-specific adipose deposition are also reported to be influenced by circLCLAT1, circFNDC3AL, circCLEC19A, and circARMH1 through the PPARs pathway. circ-PLXNA1 is primarily expressed in the liver and adipose tissue of ducks. The development of duck adipose cells is restricted by circ-PLXNA1 inhibition [218]. Studies have also observed and predicted that a possible target for the therapy of hepatic steatosis is circRNA 0046366 (Table 2) [161].

## 6. Role of lincRNA Paral1 in Adipogenesis and Activation of PPARγ

Adipocyte differentiation and functioning depend on a complex network of interrelated transcription factors focused on PPARγ [220]. lincRNA promotes adipocyte development and coactivates the master adipogenic regulator PPARγ via association with the paraspeckle component and hnRNP-like RNA binding protein 14 (RBM14/NCoAA); hence, it was named PPARγ-activator RNA-Binding Motif Protein 14 (RBM14)-associated lncRNA (Paral1) [221,222]. According to a study by Firmin et al. (2017), the expression of Paral1 and PPARγ rises simultaneously during adipocyte development [222]. The presence of a PPARγ binding site upstream of the Paral1 transcription start site indicates that PPARγ may regulate the production of a coactivating RNA. It would create a favorable feedforward loop on PPARγ expression, prompting the inquiry into the effect of PPARγ agonism on the expression of Paral1. The molecular mechanism for Paral1′s pro-adipogenic action is its ability to engage with and enhance the coactivating capacity of RBM14, whose expression tracks Paral1′s throughout adipocyte maturation. This newly found function of RBM14 is likely the result of its capacity to operate as an indirect coactivator via synergistic effects with nuclear receptor coactivators. The expression of Paral1 is confined to adipocytes and decreases with a rising body mass index in humans [223,224]. 

## 7. Future Perspectives

The interaction of pharmacologic drugs with PPARs’ modulation resulting from the environment of the individual patients is a facet of PPARs-targeted therapy that requires additional research. Lifestyle modifications may have an immediate impact on PPARs-directed environmental parameters. Numerous metabolic processes are modulated by PPAR**s**, which also cause pleiotropic effects in numerous tissues [225]. Specialized effects can be evoked and combined using compounds with different activity patterns on various PPAR**s** isotypes, further adding to the complexity [226]. It offers significant prospects and problems for studying PPAR treatments for NAFLD.

## 8. Conclusions

Several ncRNAs have been involved in the etiology of NASH, making them attractive targets for RNA-based therapies. Antisense oligonucleotides (ASOs) are typically 14–20 single-stranded nucleotides intended to bind and suppress complementary RNA transcripts [227,228]. Most ASOs have phosphorothioate (PS) backbone linkages to improve their pharmacokinetic characteristics in vivo [227,228]. The FDA-approved ASO-based drug eteplirsen for the therapy of Duchenne muscular dystrophy [229] and nusinersen for diagnosing the neurodegenerative disease spinal muscular atrophy in both cases [230] demonstrate the high clinical efficacy of chemically altered ASOs as RNA-targeted therapeutics [231]. 

In order to distinguish between the various stages of NAFLD, ncRNAs are emerging as helpful biomarkers. The majority of clinical development will be concentrated on drugs that bind to miRNAs or have an impact on how splicing functions. Our knowledge of how lncRNAs and other ncRNAs function will probably improve over time. New categories of possible non-coding targets might also appear. The design of efficient development programs will gain a more robust foundation as our knowledge of ncRNAs, and their mechanisms improves, and the likelihood of clinical success will increase [232]. 

Today, the prevalence of NAFLD and NASH is rising, which correlates favorably with the rate of obesity and diabetes. NAFLD and NASH are the leading contributors to the advancement of HCC, the predominant form of liver cancer. Furthermore, there is no effective cure for NAFLD and NASH. For the future diagnosis of NAFLD, new noninvasive diagnostic markers such as miRNAs have been investigated. Since PPARs are one of the primary regulators and targets of ncRNA, several studies have focused on the treatment strategies for NAFLD via ncRNA-PPARs regulation. As the number of lncRNAs is greater than that of protein-coding genes, a more focused, detailed system of lncRNAs, mainly using genetic manipulation and knockdown by ASOs, is necessary to verify prior reports on lncRNAs in the liver-directed at elucidating their physiological functions and to mentor new treatment strategies to fight life-threatening liver diseases, such as NASH. The rising significance of ncRNAs as therapeutic agents constitutes a landmark in developing innovative treatment techniques. The fascinating field of lncRNA is still in its infancy, and additional experimental research is required to comprehend their potential as therapeutic targets. Understanding the cellular circuits and network-associated miRNA necessitates comprehending the various miRNA processes, including the decoy function and 5′UTR regulatory activity.

## Figures and Tables

**Figure 1 cells-11-03959-f001:**
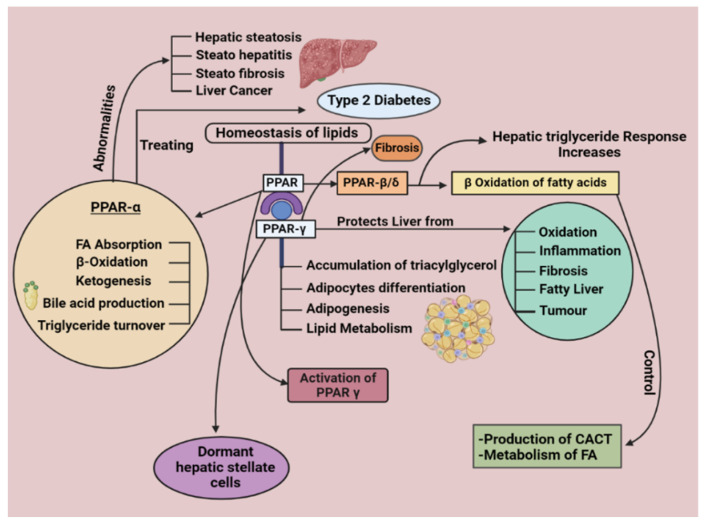
This illustration represents the regulatory mechanisms of PPARs involved in NAFLD. It gives a complete overview of the various PPARs (PPARα, PPARγ, and PPARβ/δ) and their critical roles and mechanisms in the body.

**Figure 2 cells-11-03959-f002:**
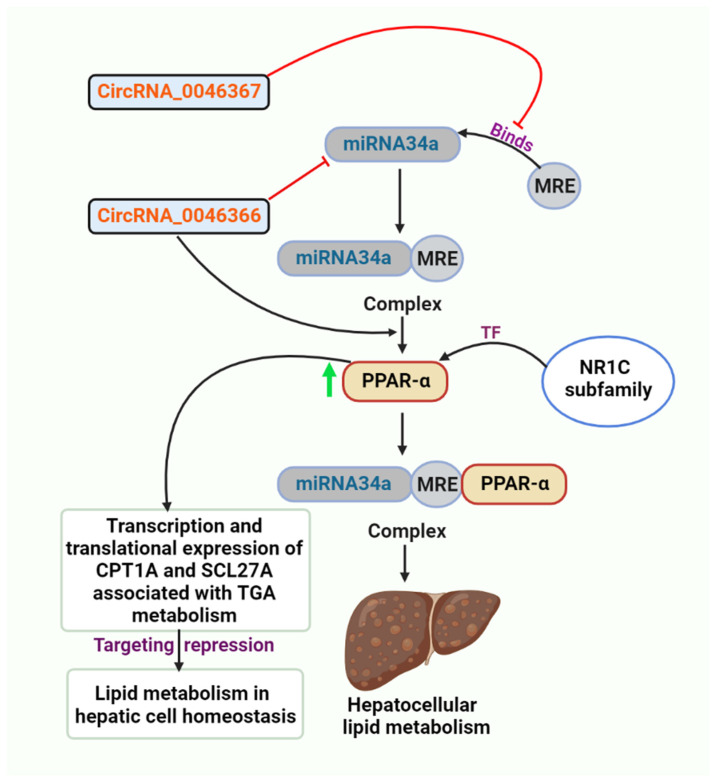
The figure depicts the inhibitory role of circRNA_0046367, preventing MRE from binding miRNA34a, and the role of circRNA_0046366 in inhibiting miRNA34a.

**Figure 3 cells-11-03959-f003:**
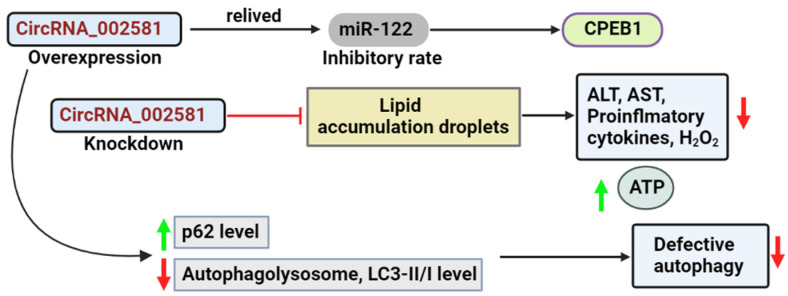
This figure portrays the result of overexpression and the knockdown of the circRNA_002581.

**Figure 4 cells-11-03959-f004:**
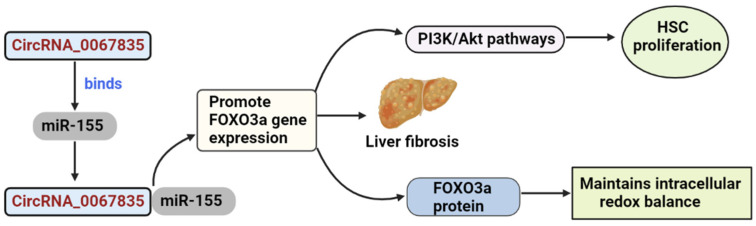
This figure depicts the role of circRNA_0067835 in promoting the FOXO3a gene expression.

**Figure 5 cells-11-03959-f005:**
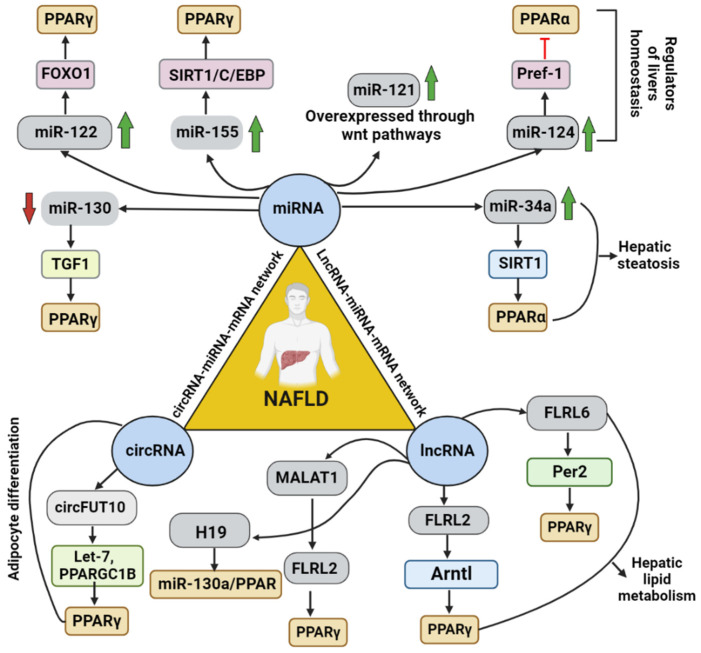
Role of ncRNAs and their respective molecular factors found to be involved in regulating NAFLD progression via targeting PPARs. This figure depicts the suspected pathway and molecules involved in regulating different ncRNA and the type of PPARs involved.

**Table 1 cells-11-03959-t001:** Table showing the research studies completed on different miRNAs and lncRNAs and the underlying mechanism.

ncRNA	Cells Involved	Animal Model Used	Genes Downregulated/Upregulated	Mechanism	Pathway Activated	Reference
miR-122	HepG2 and Huh-7 cells	16 C57BL/6 mice (males, 8 weeks old)	Downregulation of Sirt1	Binds to 3′-UTR and substantial mRNA degradation downregulates sirt1.	AMPK pathway	[146]
miR-21	CD45^+^ cells, T cells	Eight-week-old male Ldlr^−/−^ mice, nine-week-old male WT or miR-21^−/−^ mice	Downregulation of ß-oxidation genes Cpt1 and Acox1	Through normalizing liver PPARα, miR-21 inhibition and suppression significantly reduced liver damage, inflammation, and fibrogenesis.	PPARα pathway	[84]
miR-34a	Kupffer cells	miR-34a^fl/fl^ mice, albumin-Cre (Alb-Cre) mice, and C57BL/6J mice	Induces (Cyp7a1 and Cyp8b1)	miR-34a promotes the induction of Kupffer cell activation/inflammation, lipotoxicity, and apoptosis; boosts CYP7A1 and CYP8B1 expression, which causes a temporary drop in free cholesterol (FC) levels and the subsequent induction of SREBP2 processing and its target genes, including HMGCR. It enhances the conversion of FC to bile acids.	miR-34a-sirtuin 1 (SIRT1)-SREBP1c pathway activates hepatic TGFb signaling,	[147]
miR-192	HepG2 cells	Male C57BL/6 mice	Downregulation of SREBF1 target genes, including FASN and PPARG i	miR192 acts directly on the 3′UTR of SREBF1, which results in the dysregulation of lipid homeostasis in the hepatocytes.	Suppress SBREPF1	[148]
lncRNA MALAT1	HepG2 and AR42J	Male C57BL/6 mice (8 weeks old, *n* = 6/group)	---------	PPARα expression was significantly increased, and CD36 levels were significantly decreased when MALAT1 was knocked down.	MALAT1 could regulate PPARα/CD36 through the mediation of the miR-206/ARNT axis	[115]
lncRNA HOTAIR	HepG2 cells	Adult (8-week-old) male C57BL/6J mice	HOTAIR knockout increases the expression of genes	HOTAIR enhances the accumulation of PRC2 and H3K27 trimethylation to the MEG3 promoter	Regulates the expression of the DNMT1/MEG3/p53 pathways	[117]
lncRNA APTR	Hepatic stellate cells	(CCl_4_)-treated mice	Attenuates TGFB1-induced upregulation of ACTA2	The overexpression of α-SMA in HSCs brought on by TGF-β1 is prevented by suppressing APTR.	Abrogate TGF-β_1_-induced upregulation of α-SMA	[69,149]
lncRNA PVT1	Primary HSCs cells	Eight-week-old male C57BL/6J mice	Regulates the expression of PTCH1, SMO, and GLI2	Chromatin modification, transcriptional regulation, and post-transcriptional regulation	PVT1 silencing inhibits the Hh pathway	[150]
lncRNA COX2	IEC cell line (IEC4.1)	C57BL/6J mice	Enhances the transcription of Il12b	By controlling Mi-2/NuRD-mediated epigenetic histone changes, LincRNA-Cox2 affects the transcription of the Il12b gene in intestinal epithelial cells stimulated by TNF.	Activates NF-ĸB signaling pathway	[126]
lncRNA NEAT 1	HepG2 cells and LO2 cells	------	Activates lipogenesis-related genes, such as ACC and FAS	By targeting miR-139-5p, NEAT1 knockdown reduces lipid buildup in the NAFLD cellular model.	Regulates the c-Jun/SREBP1c pathway	[151]
lncRNA UC372	HepG2 cells	C57BL/6J mice	Regulates the expression of genes related to lipid synthesis and uptake, including ACC, FAS, SCD1, and CD36	By inhibiting miR-195/miR-4668 maturation from reversing miR-195/miR-4668-mediated suppression of functional target gene expression, uc.372 promotes hepatic steatosis.	-----	[135]
lncARSR	human HCC cells (HepG2)	C57BL/6 male mice (aged 6 weeks)	-------	lncARSR inhibits YAP1, which inhibits phosphorylation nuclear translocation	Suppresses IRS2/AKT pathway	[31]
lncRNA APOA4-AS	Primary hepatocytes cells	C57BL/6 mice	Decreases APOA4 mRNA and protein levels, whereas expression of other lipid metabolism-associated genes (e.g., FASN, SCD1, APOB, MTP, CPT1α, and MCAD) were not altered	HuR, a protein that stabilizes mRNA, and APOA4-AS interact to stabilize APOA4 mRNA.	Regulates many metabolic pathways	[76]
lncRNA H19	HepG2 and Huh-7 cells	C57BL/6 mice (males, 8 weeks old)	H19 and PPARγ were upregulated	H19 exerts its biological functions in NAFLD by using a ceRNA mechanism to control the expression of miR-130a’s downstream genes.	Regulates miR-130a/PPARγ pathway	[152]

**Table 2 cells-11-03959-t002:** Pathway and proteins involved in PPARs-related ncRNA in NAFLD and the possible treatment strategy.

ncRNA	Expression Pattern	Type of PPAR	PPARs Associated Molecular Factor	Type of Disease	Possible Treatment and Advantages	References
MiR-124	Elevated	PPARα	Pref 1, Notch signaling,	NAFLD—Hepatic lipid metabolism	Antisense miR-124 can be used. MiR-124 acts as a biomarker in NAFLD progression	[192]
MiR-21	Elevated	PPARα	LRP6, WNT/β-catenin signaling, SFRP5 pathway	Liver steatosis	Antisense MiR-21 therapy and biomarker for early diagnosis	[116]
MiR-122	Elevated	PPARγ	FoxO1	Hepatic cholesterol	Antisense MiR-122, which inhibits the production	[203]
MiR-34a	Elevated	PPARα	SIRT1, AMPK activation	Hepatic steatosis	Antisense MiR-34a	[120]
MiR-130	Reduced	PPARγ	TGF-1	Fibrotic steatohepatitis, insulin signaling	MiR-130 can be administered	[206]
MiR-155	Elevated	PPARγ	C/EBP, SIRT1	Adipogenesis	Antisense-155	[219]
circFUT10	Elevated	PPARγ	Let-7, PPARGC1B	Adipocyte Differentiation	--	[217]
circRNA_0046367/circRNA_0046366	Reduced	PPARα	CPT2, ACBD3, SLC27A	Hepatic steatosis	circRNA_0046367/circRNA_0046366	[158]
lncRNA FLRL6/FLRL2	Elevated	PPARγ PPARγ	circadian rhythm- Per2, Arntl	Hepatic lipid metabolism	--	[158]
LncRNA HI9	Elevated	PPARγ	miR-130a/PPAR axis	Liver fibrosis	--	[152]
MALAT 1	Elevated	PPARα	miR-206/ARNT	Liver fibrosis	--	[115]

## Data Availability

Data are available from the authors on request (AVG).

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
