# Peer review of "Exploring the Regulatory Role of ncRNA in NAFLD: A Particular Focus on PPARs"

_cells, 2022, doi:10.3390/cells11243959_

Round 1

Reviewer 1 Report

The authors made a very comprehensive review of the role of ncRNAs in NAFLD development, with particular focus on role of ncRNAs and PPARs. Overall, the work is well structured and easy to follow, with detailed diagrams and table guide the reader through the manuscript.

Given that global roles of ncRNAs in NAFLD (whether they regulate PPARs or not) are also included in the review I would suggest to change the title to reflect it, so the readers know they will found a very detailes review on ncRNAs in NAFLD. I would suggest something like "Exploring the Regulatory Role of ncRNA in NAFLD: a particular focus on PPARs".

In section 4.1 I would suggest to include a last section 4.1.6 about other miRNAs that have been related with NAFLD/NASH and lipid metabolism (e.g. hepatic miR-33, miR-29, miR-873 or miR-221/222). 

As minor correction I noted that through the abstract PPARs are reffered in singular for the construction of the sentences (example: PPARs plays, PPARs was found), while the rest of the text they are in plural. I think they should be always referred in plural.

2. Lines57-58. Please rewrite the sentences, is not clear what the authors mean.

3. lines 65-67. the authors refers twice to PPARg in adipocytes and liver expression. 

Finally, in the conclusion, I would include a little description of the potential controversy of undesired effect of targeting ncRNAs due to offtarget gene or tissue effects, and how this could be overcome.

Author Response

The authors made a very comprehensive review of the role of ncRNAs in NAFLD development, with particular focus on role of ncRNAs and PPARs. Overall, the work is well structured and easy to follow, with detailed diagrams and table guide the reader through the manuscript.

Given that global roles of ncRNAs in NAFLD (whether they regulate PPARs or not) are also included in the review I would suggest to change the title to reflect it, so the readers know they will found a very detailes review on ncRNAs in NAFLD. I would suggest something like "Exploring the Regulatory Role of ncRNA in NAFLD: a particular focus on PPARs".

Response: All the authors are thankful to the reviewer for evaluating the manuscript data and potential comments for enhancing the quality of the manuscript. We have changed the Title as per the suggestion. The track changes are made in the revised manuscript.

In section 4.1 I would suggest to include a last section 4.1.6 about other miRNAs that have been related with NAFLD/NASH and lipid metabolism (e.g. hepatic miR-33, miR-29, miR-873 or miR-221/222). 

Response: We have added the 4.1.6 subsection as per the suggestion. The track changes are made in the revised manuscript.

As minor correction I noted that through the abstract PPARs are reffered in singular for the construction of the sentences (example: PPARs plays, PPARs was found), while the rest of the text they are in plural. I think they should be always referred in plural.

Response: We have made the necessary changes in the revised file as per the suggestion. The track changes are made in the revised manuscript.

2. Lines57-58. Please rewrite the sentences, is not clear what the authors mean.

Response: The confusing line has been removed to avoid confusion. The track changes are made in the revised manuscript.

3. lines 65-67. the authors refers twice to PPARg in adipocytes and liver expression. 

Response: We have reframed the line as per the suggestion to avoid confusion. The track changes are made in the revised manuscript.

Finally, in the conclusion, I would include a little description of the potential controversy of undesired effect of targeting ncRNAs due to offtarget gene or tissue effects, and how this could be overcome.

Response: We have added the data as per the suggestion and tried to satisfy the comment in the conclusion section. The track changes are made in the revised manuscript.

Reviewer 2 Report

The authors wrote a very interesting revision that describes the role of the principal non-coding RNA (ncRNA) described in the pathophysiology of NAFLD. Overall the revision is well designed and list the most relevant ncRNA found in the literature. Importantly, this revision has been focused on describing the ncRNA with a relevant role in regulating lipid metabolism in liver cells in the context of NAFLD.

NAFLD is characterized by fatty acid deposition within the hepatocytes that promote lipotoxicity activating inflammatory but  also by fibrogenesis responses leading to scar deposition. Indeed, the fibrosis stage has been identified as the most important prognostic factor in patients with NAFLD.  It is important to note that patients with NAFLD decrease the ammount of fibrosis after loss of weigth. For this reason, I suggest to include in this revision a description of ncRNA related aproaches focused on fibrosis regression in NAFLD.

I would propose to include miRNA involved in the HSC activation. Actually multiple miRNAs related HSC activation have been reported and their potential anti-fibrotic effects have been shown by inhibiting in vivo and in vitro their function. Some examples are: 1) miR-21, widely described to be up-regulated in human and murine fibrotic liver tissues; 2) miR-150 and miR-194 that have been reported to play an anti-fibrogenic role 3) miR-192 found enriched in HSC with quiescent phenotype and having a role in inhibiting HSC activation.

Author Response

The authors wrote a very interesting revision that describes the role of the principal non-coding RNA (ncRNA) described in the pathophysiology of NAFLD. Overall the revision is well designed and list the most relevant ncRNA found in the literature. Importantly, this revision has been focused on describing the ncRNA with a relevant role in regulating lipid metabolism in liver cells in the context of NAFLD.

Response: Authors are thankful to the reviewer for evaluating the quality of the manuscript and potential comments for enhancing the quality of the manuscript. We have tried to satisfy all the comments throughout the revised manuscript. The track changes are made in the main manuscript.

NAFLD is characterized by fatty acid deposition within the hepatocytes that promote lipotoxicity activating inflammatory but also by fibrogenesis responses leading to scar deposition. Indeed, the fibrosis stage has been identified as the most important prognostic factor in patients with NAFLD.  It is important to note that patients with NAFLD decrease the ammount of fibrosis after loss of weigth. For this reason, I suggest to include in this revision a description of ncRNA related aproaches focused on fibrosis regression in NAFLD.

Response: We have included the data stating ncRNA-related approaches focused on fibrosis regression in NAFLD in section 4. ncRNA regulation in NAFLD. The track changes are made in the revised manuscript.

I would propose to include miRNA involved in the HSC activation. Actually multiple miRNAs related HSC activation have been reported and their potential anti-fibrotic effects have been shown by inhibiting in vivo and in vitro their function. Some examples are: 1) miR-21, widely described to be up-regulated in human and murine fibrotic liver tissues; 2) miR-150 and miR-194 that have been reported to play an anti-fibrogenic role 3) miR-192 found enriched in HSC with quiescent phenotype and having a role in inhibiting HSC activation.

Response: We have included the data stating miRNA involved in the HSC activation in subsection 4.1.6 Other miRNAs. The track changes are made in the revised manuscript.

Reviewer 3 Report

The manuscript was written by Mukherjee et al. entitled Exploring the Regulatory Role of ncRNA in NAFLD through PPARs”

Researchers suggested that in this review, the role PPARs are involved in the ncRNA pathway regulation in the formation of NAFLD. The role of ncRNAs in fatty liver and hepatic steatosis has attracted much attention during the last decade. There is no certain cure for NAFLD, and routine treatments are a low-fat diet, weight loss, and diabetes control. In terms of treatment, there is also a possibility to target ncRNAs for therapeutic approaches. The inhibition or mimicking of ncRNAs is one of the promising approaches in NAFLD's targeted therapy.

This is a narrative review of the various molecular and genetic drivers of NAFLD that can be considered as potential contributing factors in combating the development of mild steatosis to severe  non-alcoholic steatohepatitis (NASH).

Currently, the manuscript requires that several of the essential paragraphs of the review be supported by bibliographical references.

In addition, in some paragraphs, the jump between one topic and another is very abrupt, so it is necessary to improve the writing.

Comments for the authors

Write a brief introduction to NAFLD and NASH disease.

Please mention the brief discussion on molecular mechanism of hepatic steatosis.

The visceral adipose tissue that produces proinflammatory cytokines (TNF-alpha, IL-6, and IL-17) and growth factors referred to as adipokines (adiponectin, omentin, chemerin) leading to dyslipidemia, insulin resistance, diabetes, cardiovascular disorders. Please mention these factors.  

Please explain the molecular level, Effect of the endocrine hormone on ncRNAs in fatty liver.

The investigation demonstrated that InsR was often expressed in GBM surgical tissues and xenograft tumor lines, with mitogenic isoform-A being the most abundant.

There are some abbreviations missing from this list.

Author Response

The manuscript was written by Mukherjee et al. entitled Exploring the Regulatory Role of ncRNA in NAFLD through PPARs”

Researchers suggested that in this review, the role PPARs are involved in the ncRNA pathway regulation in the formation of NAFLD. The role of ncRNAs in fatty liver and hepatic steatosis has attracted much attention during the last decade. There is no certain cure for NAFLD, and routine treatments are a low-fat diet, weight loss, and diabetes control. In terms of treatment, there is also a possibility to target ncRNAs for therapeutic approaches. The inhibition or mimicking of ncRNAs is one of the promising approaches in NAFLD's targeted therapy.

This is a narrative review of the various molecular and genetic drivers of NAFLD that can be considered as potential contributing factors in combating the development of mild steatosis to severe  non-alcoholic steatohepatitis (NASH).

Currently, the manuscript requires that several of the essential paragraphs of the review be supported by bibliographical references.

In addition, in some paragraphs, the jump between one topic and another is very abrupt, so it is necessary to improve the writing.

Response: Authors are thankful to the reviewer for evaluating the quality of the manuscript and potential comments for enhancing the quality of the manuscript. We have tried to satisfy all the comments throughout the revised manuscript. The track changes are made in the main manuscript.

Comments for the authors

Write a brief introduction to NAFLD and NASH disease.

Response: We have given the data about NAFLD and NASH disease in the introduction section. The track changes are made in the main manuscript.

Please mention the brief discussion on molecular mechanism of hepatic steatosis.

Response: We have given the data about the molecular mechanism of hepatic steatosis in the introduction section. The track changes are made in the main manuscript.

The visceral adipose tissue that produces proinflammatory cytokines (TNF-alpha, IL-6, and IL-17) and growth factors referred to as adipokines (adiponectin, omentin, chemerin) leading to dyslipidemia, insulin resistance, diabetes, cardiovascular disorders. Please mention these factors.  

Response: We have mentioned these factors in the introduction section. The track changes are made in the main manuscript.

Please explain the molecular level, Effect of the endocrine hormone on ncRNAs in fatty liver.

The investigation demonstrated that InsR was often expressed in GBM surgical tissues and xenograft tumor lines, with mitogenic isoform-A being the most abundant.

Response: We have added the required data in the introduction section. The track changes are made in the main manuscript.

There are some abbreviations missing from this list.

Response: We have made the changes and added the full form of the missing abbreviation. The track changes are made in the main manuscript.

Reviewer 4 Report

The paper submitted by Mukherjee et al. had the main focus to review the regulation of PPARs through ncRNAs and their role in NAFLD. The subject of the review is very interesting, and currently very "hot" since liver diseases associated with metabolic stress are very relevant subjects in the literature. The review is very broad and involves terms such as PPARs and their regulation by ncRNAs. However, while reading, some points caught my attention.

As a suggestion, the authors should consider using the more appropriate and recent term MAFLD (metabolic associated fatty liver disease, DOI: 10.1016/j.dld.2020.09.013) instead of NAFLD, which is a deprecated term.

The statement in line 90 that PPAR-gamma is the most abundant isoform in the liver should be revised.

Several times in the text I missed an introduction to the topic of miRNAs, just a basic introduction to their structure and function.

I also missed the review of an introduction to the topic of lncRNAs.

In general, the review is very long and presents themes that were not properly introduced. Although the authors intended to relate ncRNAs to PPARs in NAFLD, there is much prior text on this central theme, which in my opinion lengthens and blurs the text.

At the same time, it lacks an introduction, even minimal on the subject of ncRNAs, indicating the differences between circRNAs, lncRNAs, and miRNAs.

Author Response

The paper submitted by Mukherjee et al. had the main focus to review the regulation of PPARs through ncRNAs and their role in NAFLD. The subject of the review is very interesting, and currently very "hot" since liver diseases associated with metabolic stress are very relevant subjects in the literature. The review is very broad and involves terms such as PPARs and their regulation by ncRNAs. However, while reading, some points caught my attention.

Response: Authors are thankful to the reviewer for evaluating the quality of the manuscript and for potential comments for enhancing the quality of the manuscript. We have tried to satisfy all the comments throughout the revised manuscript. The track changes are made in the main manuscript.

As a suggestion, the authors should consider using the more appropriate and recent term MAFLD (metabolic associated fatty liver disease, DOI: 10.1016/j.dld.2020.09.013) instead of NAFLD, which is a deprecated term.

Response: The authors thank the reviewer for the valuable comment. To prevent confusion, we have used the term NAFLD instead of MAFLD, as all the cited references relate to NAFLD and not MAFLD.

The statement in line 90 that PPAR-gamma is the most abundant isoform in the liver should be revised.

Response: We have revised the suggested line as Adipose tissue is where PPARγ is highly expressed, and it plays a crucial part in controlling adipocyte differentiation, adipogenesis, and lipid metabolism. The track changes are made in the revised manuscript.

Several times in the text I missed an introduction to the topic of miRNAs, just a basic introduction to their structure and function.

Response: We have given a separate paragraph in the introduction section stating the structure and function of miRNAs and lncRNAs. The track changes are made in the revised manuscript.

I also missed the review of an introduction to the topic of lncRNAs.

Response: We have given a separate paragraph in the introduction section stating the structure and function of miRNAs and lncRNAs. The track changes are made in the revised manuscript.

In general, the review is very long and presents themes that were not properly introduced. Although the authors intended to relate ncRNAs to PPARs in NAFLD, there is much prior text on this central theme, which in my opinion lengthens and blurs the text.

Response: The authors are thankful to the comment. As per the suggestion, we have added the data in section 5. ncRNA regulation of PPARs in NAFLD, and also tried to modify the data to maintain the quality of the manuscript. The track changes are made in the revised manuscript.

At the same time, it lacks an introduction, even minimal on the subject of ncRNAs, indicating the differences between circRNAs, lncRNAs, and miRNAs.

Response: We have tried to satisfy the comment and given a separate paragraph in the introduction section on different ncRNAs. The track changes are made in the revised manuscript

Round 2

Reviewer 3 Report

Well improved, please accept it